# Emotional Intelligence and Burnout among Adolescent Basketball Players: The Mediating Effect of Emotional Labor

**DOI:** 10.3390/sports12100266

**Published:** 2024-09-29

**Authors:** Wenjun Xue, Yiming Tao, Yangyi Huang, Guannan Liu, Huiru Wang

**Affiliations:** Department of Physical Education, Shanghai Jiao Tong University, Shanghai 200240, China; xwj2026@sjtu.edu.cn (W.X.); maxim0606@sjtu.edu.cn (Y.T.); hyy2026@sjtu.edu.cn (Y.H.); guannan2020@sjtu.edu.cn (G.L.)

**Keywords:** adolescent athletes, emotional intelligence, burnout, emotional labor, basketball players

## Abstract

Burnout, characterized by emotional and physical exhaustion, poses a significant challenge to adolescent athletes, particularly in high-intensity sports like basketball. Emotional Intelligence (EI) is the ability to manage emotions, which is negatively associated with burnout. Emotional labor, including strategies of surface acting (SA), deep acting (DA), and genuine expression (GE), plays a potentially key role in emotion management between EI and burnout for athletes. This study aims to investigate the relationship between EI and burnout, as well as the mediating role of emotional labor strategies among adolescent basketball players. Our cross-sectional study, conducted in youth sports schools in four different places in China, involved 260 basketball players. Results indicate a negative association between EI and burnout, with SA and GE emerging as significant mediators. SA was positively linked to burnout, while GE showed a negative association. These findings suggest that enhancing EI and managing emotional labor strategies are crucial for mitigating burnout and improving the well-being and performance of young athletes.

## 1. Introduction

Burnout, characterized by emotional and physical exhaustion, is an increasingly critical concern for adolescent athletes facing the dual challenges of sports development and personal growth [1,2]. Characterized by a range of symptoms, including a reduced sense of accomplishment, physical and emotional exhaustion, and sport devaluation, burnout can erode the motivation, performance, and overall enjoyment of athletes [1]. The syndrome not only affects the individual itself but can also reverberate through their support systems, including coaches, families, and teammates. The consequences of athlete burnout are far-reaching, leading to issues such as decreased performance, increased risk of injury, early attrition in youth, and even a direct threat to the goal of lifelong physical activity and the wide-ranging health benefits [3,4]. The growing professionalism of youth sports has raised concerns about the early specialization and intense training schedules that young athletes face. This trend is often blamed for the heightened risk of overuse injuries, overtraining, and especially athlete burnout. The pressure to excel early, as well as the belief that early specialization is crucial for future sports success, may lead to excessive training loads [4]. This, in turn, may diminish the athletes’ intrinsic motivation and pleasure derived from sports participation, leading to a paradoxical increase in burnout and dropout rates.

Athletes need to motivate themselves to achieve long-term goals through hard training when playing sports [5]. They must consistently manage the stress of intense training and competition, which includes understanding and regulating their own emotions as well as those of others (such as teammates, opponents, coaches, referees, and spectators) [6]. Among the factors affecting an athlete’s burnout, emotional intelligence (EI) is a notable predictor. EI, first proposed by Salovey and Mayer in 1990, encompasses the abilities to accurately perceive, express, and regulate emotions and to utilize them to motivate, plan, and achieve in life [7]. Mayer described the concept of EI as follows, “the ability to perceive accurately, appraise, and express emotion; the ability to access and/or generate feelings when they facilitate thought; the ability to understand emotion and emotional knowledge; and the ability to regulate emotions to promote emotional and intellectual growth” [8]. The Four-Branch Model of EI includes four areas: (a) accurately perceiving emotion, (b) using emotions to facilitate thought, (c) understanding emotion, and (d) managing emotion [8,9]. EI is a psychological skill in the field of sports that influences emotional control by athletes, decision-making, and sporting performance itself [10]. Recent studies have begun to explore the significance of EI in sports, highlighting its role in enhancing performance and coping with the pressures inherent to competitive environments [11]. Athletes with higher EI are often better equipped to handle stress, maintain motivation, gain achievements, and foster positive relationships with teammates and coaches [12,13,14]. These attributes are crucial in the high-stakes environment of team sports based on communication and coordination, like basketball. Additionally, the prevalence of burnout among athletes has been linked to overtraining and excessive competition pressure, which can undermine an athlete’s motivation and performance [15]. These studies underscore the importance of understanding how EI can serve as a buffer against burnout.

Although previous studies have examined the link between EI and burnout, potential mediating factors remain insufficiently explored. Emotional labor, which involves managing feelings and expressions to meet professional emotional demands during interactions with customers, colleagues, and supervisors [16], could be one such factor that may play a crucial role in this context. Emotional labor strategies include surface acting (SA), deep acting (DA), and genuine expression (GE) [17]. SA involves modifying visible emotional expressions without altering internal feelings, typically requiring the suppression or feigning of emotions to fulfill external expectations [18]. DA, conversely, is proactive, involving the regulation of internal emotions to match desired expressions, leading to a more genuine emotional experience [19]. GE refers to the spontaneous expression of emotions that are naturally felt and considered appropriate for the situation [20]. These strategies highlight the dynamic nature of emotion regulation in various professional and performance-oriented settings. Emotional labor is a critical component of athletic performance and well-being, influencing how athletes manage the emotional demands of sports [21]. Different from adult professionals, adolescent athletes develop both athletic skills and emotional and social competencies. Compared to other environments, the emotional labor required in high-stakes team sports, such as managing one’s emotions during a game or interacting with teammates and coaches, may have a different impact on burnout. Given the rapid physical and emotional changes during adolescence, the interplay between EI, emotional labor, and burnout might be particularly noteworthy.

According to the Conservation of Resources (COR) theory developed by Hobfoll [22,23], individuals strive to obtain, retain, and protect resources that they value. Stress occurs when individuals face resource loss or resource threats, and it can be exacerbated when individuals invest resources without gaining expected returns [24,25]. EI is posited as a pivotal personal resource, equipping individuals with the capacity to adeptly navigate and regulate emotional terrains. The efforts required for emotional labor, especially SA, can deplete personal resources. This is achieved by necessitating the suppression or feigning of emotions, which can precipitate emotional exhaustion and escalate the incidence of burnout [26]. High EI, which enhances emotional management skills, is expected to reduce the emotional dissonance associated with emotional labor demands, conserving personal resources and reducing burnout risk. Burnout manifests as emotional exhaustion, conceptualized as a manifestation of resource depletion. This indicates that the resources consumed in emotional labor are not adequately replenished, ultimately leading to increased stress and decreased mental health [27]. Individuals with higher EI are expected to employ effective emotional labor strategies, leading to lower burnout. Conversely, those with lower EI are projected to adopt less effective emotional labor strategies, culminating in higher burnout.

Emotional labor has been demonstrated to correlate with both EI and burnout. Research indicates that the antecedents of emotional labor strategies may encompass positive and negative emotions, performance rules, customer orientation, and the alignment of emotional demands with abilities primarily influenced by EI [28]. Numerous studies have shown a distinct correlation between EI and emotional labor strategies [29,30,31,32]. In sports, EI and burnout are recognized as pivotal factors affecting athletes’ performance and well-being. A study by Ha et al. investigated the link between emotional labor strategies and burnout among South Korean sports coaches, emphasizing the moderating effect of social support [33]. This is consistent with the broader understanding that EI is a significant predictor of effective emotional management and a crucial element in reducing burnout. Furthermore, Yu and Cheng’s (2024) research on elite sports coaches in Sichuan Province, China, revealed the complex interplay between job stress, occupational burnout, coping strategies, and organizational support, underscoring the mediating role of perceived organizational support [34]. Additionally, the conceptual model proposed by Lee, Chelladurai, and Kim (2015) offers a framework for understanding emotional labor in sports coaching, identifying SA, DA, and GE as the main types of emotional labor, and linking them to job satisfaction and burnout through the psychological costs associated with each strategy [35].

This study addresses a critical gap in the literature by examining the mediating role of emotional labor strategies on the relationship between EI and burnout among adolescent basketball players. The increasing professionalization and intensity of youth sports training underscore the need to understand factors that contribute to or protect against burnout. Our research aims to provide a nuanced understanding of the emotional challenges faced by young athletes and to identify strategies that could support their emotional well-being and prevent burnout. Guided by COR theory, our hypotheses are as follows:

**H1:** 
*Higher levels of EI will be associated with lower levels of burnout among adolescent basketball players, indicating the potential of EI as a protective factor.*


**H2:** 
*The use of SA, DA, and GE in emotional labor will mediate the relationship between EI and burnout, suggesting that how athletes manage their emotions significantly influences their susceptibility to burnout.*


## 2. Materials and Methods

### 2.1. Participants

The study utilized a convenience sampling method, targeting female basketball players from the first year of middle school to the third year of high school. Participants were recruited from four sports schools in China, located in Shanghai, Chongqing, Fujian, and Shandong provinces, with data collection occurring between March and April 2024. The schools were chosen for their diverse regional representation and the presence of a significant number of adolescent female basketball players. Including all eligible athletes from these schools facilitated a thorough examination of the study objectives, eliminating the need for random selection and ensuring the sample’s accessibility and relevance to the research questions. Data was collected from a total of 277 athletes from professional basketball teams. Athletes were briefed on the research procedures and provided informed consent prior to survey administration. Participants were volunteers and received no compensation for their involvement. Data was initially screened to identify and exclude invalid questionnaires based on predefined criteria. Criteria included completion times within the 5th to 95th percentiles of the overall distribution, with outliers suggesting potential inattention or disengagement. Furthermore, questionnaires with consistently identical responses were reviewed to ensure thoughtful engagement with the survey content; those lacking this were excluded. Inconsistencies in responses to reverse-scored items were assessed, with significant deviations indicating potential misunderstandings or misinterpretations, leading to exclusion. Finally, any incomplete questionnaires were deemed invalid. Applying these standards rigorously, 17 questionnaires were identified as invalid and excluded, resulting in a final dataset of 260 valid responses.

### 2.2. Measurements

Participants completed a questionnaire that included the Wong and Law Emotional Intelligence Scale, a composite scale assessing emotional labor, the Athlete Burnout Questionnaire, and items regarding gender, age, years of training, and national athletic certification level.

#### 2.2.1. Emotional Intelligence

EI was measured using the Wong and Law Emotional Intelligence Scale (WLEIS) [36]. According to the definition by Mayer and Salovey, this scale covers the ability to understand one’s own emotions, understand the emotions of others, regulate one’s own emotions, and use one’s own emotions [37]. The WLEIS consists of 16 items, including ‘I really understand what I feel’ and ‘I have good control of my own emotions’ with four subscales corresponding to the four components of EI: self-emotional appraisal, others’ emotional appraisal, regulation of emotion, and use of emotion [38]. WLEIS items are rated on a 7-point Likert scale, from 1 (totally disagree) to 7 (totally agree). Higher scores on the scale indicate higher levels of EI. The Chinese version of the WLEIS demonstrated adequate internal consistency and validity in Chinese university students and youth populations [39,40]. The Cronbach’s alpha for the total WLEIS was 0.916 in this study.

#### 2.2.2. Emotional Labor

A composite scale was used to assess emotional labor, integrating various strategies athletes use to manage emotions in high-pressure sports settings. This composite scale encompasses three distinct components: SA, DA, and GE. Each component is evaluated using a different set of items to capture the nuanced ways in which athletes regulate their emotions. SA and DA were examined using the Emotional Labor Scale by Brotheridge and Lee [41], while GE was examined using a three-item scale based on the measures of expression of naturally felt emotions developed by Diefendorff et al. [16].

SA: This aspect of emotional labor involves the alteration of visible emotional expressions without changing internal feelings. It often requires the suppression or feigning of emotions to meet external demands. The scale for SA includes items such as “Hide my true feelings about a situation”, measured on a 5-point Likert scale. A previous study reported a Cronbach’s alpha of 0.86 for SA.

DA: Distinct from SA, DA is proactive and involves regulating internal emotions to align with desired expressions. This strategy reflects a more authentic emotional experience and is assessed with items like “Make an effort to actually feel the emotions that I need to display to others”. Brotheridge and Lee reported a Cronbach’s alpha of 0.89 for DA.

GE: This component represents the spontaneous and unconstrained display of emotions that are naturally felt and deemed appropriate in each context. The scale for GE is based on the measures of expression of naturally felt emotions, with items such as “I express genuine emotions to athletes”, also rated on a 5-point Likert scale.

The Cronbach’s alpha of the SA, DA, and GE subscales and the total scale were 0.809, 0.774, 0.714, and 0.793 in this study, respectively. Utilizing a composite scale provides a holistic understanding of the emotional labor strategies used by adolescent basketball players, reflecting the complexity of emotional regulation in sports. The clarity and comprehensiveness of this scale are crucial for accurately assessing the mediating role of emotional labor strategies between EI and burnout.

#### 2.2.3. Burnout

The athlete Burnout Questionnaire (ABQ) was used to measure participants’ burnout [1]. The ABQ includes 15 items measuring emotional/physical exhaustion (5 items), reduced sense of accomplishment (5 items), and sport devaluation (5 items). Responses were measured on a 5-point Likert scale ranging from 1 (never) to 5 (always). Previous research has consistently revealed that the ABQ had satisfactory validity and reliability among Chinese collegiate athletes [42,43]. The Cronbach’s alpha for the total ABQ was 0.861 in our study.

### 2.3. Statistics and Analysis

To rigorously examine the mediating role of emotional labor strategies on the relationship between EI and burnout, a series of regression analyses were conducted using SPSS 24.0, complemented by the PROCESS v4.3 macro developed by Hayes (Mesquite, TX, USA). This approach facilitated the assessment of multiple mediators within a single analytical framework. The mediating model was carefully constructed to evaluate the indirect effects of EI on burnout through the emotional labor strategies of SA, DA, and GE.

The analysis commenced with descriptive statistics to outline the central tendencies and variability of the variables. This was followed by a correlation analysis to establish the preliminary relationships between EI, emotional labor strategies, and burnout. Subsequently, hierarchical regression analyses were performed to determine the unique contributions of EI and emotional labor strategies to burnout, with control variables such as gender, age, years of training, and athletic level initially entering the model.

The PROCESS macro was then employed to test the mediating effect of the emotional labor strategies. EI was specified as the independent variable, burnout as the dependent variable, and SA, DA, and GE as the mediators. This model allowed for the estimation of the indirect effects of EI on burnout through each emotional labor strategy, as well as the direct effect of EI on burnout. The significance of these indirect effects was determined using bootstrap confidence intervals, providing a robust test of the mediating hypotheses.

Structural equation modeling (SEM) was conducted using AMOS to validate the mediating model further. The overall fit of the model was evaluated using common fit indices, including the Comparative Fit Index (CFI), Tucker-Lewis Index (TLI), Root Mean Square Error of Approximation (RMSEA), and Standardized Root Mean Square Residual (SRMR). These indices provided a comprehensive assessment of how well the proposed model represented the data, with acceptable values indicating a good fit. This comprehensive analytical approach ensures a thorough and transparent examination of the mediating role of emotional labor strategies in the relationship between EI and burnout.

## 3. Results

### 3.1. Demographic Characteristics

Descriptive statistics on gender, age, years of training, and athletics level are presented in Table 1 by means and standard deviations (M ± SD) for continuous variables or numbers and percentages (%) for categorical variables. Among the 260 participants aged 15.1 ± 1.95, 70 (26.9%) are males. The years of training for athlete participants range from 1 year to 10 years, with an average of 3.93 ± 2.48 years. Nearly half of the athlete participants have obtained a national first-class or second-class athletic level, while the other athlete participants have not yet obtained athletic level certification due to the limitation of age or training years.

### 3.2. Common Method Bias Test

To thoroughly assess the presence of common method bias in our study, we initially employed Harman’s single-factor test, which involved conducting a non-rotating principal component analysis on all variables. The test results showed that there are 9 factors with eigenvalues greater than 1. The contribution rate of the first factor was 28.11%, far lower than the critical value of 40% [44], suggesting no serious common method bias.

We conducted a confirmatory factor analysis (CFA) to further evaluate the presence of common method variance. In this analysis, we specified a model where all items were loaded onto a single common factor, hypothesizing that if common method bias were a significant issue, this model would show a good fit to the data. The fit indices for this model were compared against those of the default model, where variables were loaded onto their respective theoretical factors.

The results of the CFA in Table 2 showed the CFI, GFI, NFI, RMR, RMSEA, and χ^2^/df of both models of default and one single common factor, indicating that the model with a single common factor did not fit the data well. Furthermore, the χ^2^/df for the model with a single common factor also increased when compared to the default model, further suggesting that the data did not support the presence of a single common method factor.

### 3.3. Correlation Analysis

The bi-variate correlations between investigated variables are presented in Table 3. There is a broad correlation between variables, which provides a reliable basis for subsequent analysis and modeling. Gender, age, years of training and athletic level are positively correlated with SA, DA, and burnout. EI is negatively correlated with gender. There is a basic correlation between EI, SA, DA, GE, and burnout.

### 3.4. Gender Difference

According to the correlation result above, we conducted an independent samples *t*-test to examine potential gender differences in EI, SA, DA, GE, and Burnout, as shown in Table 4. The analysis revealed significant differences in all variables except GE. The results indicated that males demonstrated significantly higher levels of EI compared to females. Conversely, females exhibited significantly higher levels of SA and DA than males. Furthermore, Burnout was found to be significantly higher in females.

### 3.5. Associations of EI and Emotional Labor Strategies with Burnout

The results of hierarchical linear regression analysis on the associations of EI and the three emotional labor strategies with burnout are presented in Table 5. Variance inflation factors (VIFs) of all variables are less than 5, suggesting that multi-collinearity was not an issue in the estimate. Model 1 showed the basic linear regression model of the covariates, with a total explaining effect of 10%. After adjusting for gender, age, years of training, and athlete level in Model 2, EI was negatively associated with burnout (β = −0.276, *p* < 0.001). EI accounted for an additional 21.4% of the variance of burnout. In Model 3, SA was positively associated with fatigue (β = 0.307, *p* < 0.001), whereas GE was negatively associated with fatigue (β = −0.252, *p* < 0.001). However, DA was not significantly associated with fatigue (β = 0.112, *p* = 0.066). These emotional labor strategies accounted for an additional 9.5% of the variance of burnout. When these emotional labor strategies were added in Model 3, the absolute value β of EI was diminished. Therefore, three dimensions of emotional labor strategies could probably become mediators in the association between EI and burnout.

### 3.6. Mediating Effect of Emotional Labor

The mediating effect of three emotional labor strategies between EI and burnout was examined using Model 4 of PROCESS, developed by Hayes. The result is shown in Table 6 and Table 7, and Figure 1. The total predictive effect of EI on burnout (c) was negatively significant. After adding three emotional labor strategies, the effects of EI on SA (a1) and GE (a3) were significant, while SA (b1) and GE (b3) also had significant effects on burnout, respectively. The direct effect of EI on burnout (c’) was still significant with three mediating variables. The indirect effect of SA and GE between EI and burnout was negative. However, DA was negatively related to EI (b1) or burnout (b2), indicating a non-significant pathway. In addition, the proportions of mediating roles of SA and GE were 14.17% and 9.66% in the total effect of EI on burnout, respectively.

## 4. Discussion

This study contributes to the emerging literature that explores the relationship between EI and burnout among adolescent basketball players. Our findings indicated that higher EI is associated with a reduced risk of burnout. Furthermore, the study investigated the mediating effects of three emotional labor strategies on the relationship between EI and burnout. Among these three strategies, SA and GE mediated the relationship between EI and burnout. SA was positively associated with burnout, whereas GE was negatively related to burnout.

Our findings support Hypothesis 1, revealing a significant inverse relationship between EI and burnout. This correlation suggests that athletes with higher EI are less likely to experience burnout, highlighting EI as a potential protective factor. Previous studies have focused on three groups, doctors, teachers, and students, in order to investigate this correlation. Consistent with our results, people with high levels of EI were more likely to have lower levels of burnout [45,46,47]. A study of 68 medical students used 4 questionnaires to measure EI and burnout, which showed that high EI was associated with decreased levels of burnout [48]. Additionally, a 5-year cohort study showed that general surgery residents’ populations generally experience high levels of burnout. The study demonstrated that each subfield of EI was negatively correlated with burnout [49].

Moreover, EI plays a key role in sports as a predictor of various forms of self-regulation, including identified, introjected, and external regulation [50]. Research indicates that athletes with higher EI tend to exhibit superior emotional self-regulation and empathy toward others (e.g., teammates and coaches) [5]. This deliberate emotional regulation can help athletes more effectively cope with the various stresses they face, thereby reducing the risk of burnout. Firstly, adolescent athletes need long periods of high-intensity repetitive training. Secondly, they must face pressure from coaches, parents, and many other aspects. Under these circumstances, they are more prone to exhibit negative emotions and even develop psychological problems such as depression and anxiety. Thirdly, they need to complete equally challenging academic tasks while fulfilling training and competition tasks. Therefore, the dual demands may heighten the risk of burnout for adolescent athletes [51]. Thus, EI can provide individuals with the ability to adapt to stressful situations and serve as a protective factor [52]. Coaches and physical education teachers, akin to athletes, regularly manage emotions and face similar challenges, such as setbacks, pressures, and motivational issues [53]. A study among female physical education trainee teachers aged 21–25 years revealed a significant negative association between EI and burnout, suggesting that those with lower EI are more prone to burnout [54]. Therefore, those with high EI have a better ability to reduce burnout through emotional regulation strategies, thereby adapting to stress.

Consistent with Hypothesis 2, our study indicated that emotional labor strategies, particularly SA and GE, significantly mediate the relationship between EI and burnout. This finding is instrumental for developing targeted interventions to help athletes manage their emotional expressions more effectively, thereby reducing the risk of burnout. Specifically, our study found that SA was positively associated with burnout, whereas GE was negatively associated with burnout. In addition, DA was not associated with burnout. Our results agree with those of a study by Kim in a group of hospital nurses, which demonstrated that increased use of SA can lead to an increase in burnout. While utilizing GE as an emotional labor strategy was associated with less burnout [55]. However, in sports, most studies on emotional labor and burnout have focused on the role of coaches. In a study of 259 secondary schools coaches in South Korea, a questionnaire was used to measure burnout across three dimensions: emotional exhaustion, depersonalization, and reduced personal accomplishment. The impact of the three emotional labor strategies on these burnout dimensions was also evaluated. The results showed that SA was positively correlated with each sub-dimension of burnout, while GE was negatively correlated with each sub-dimension of burnout [54]. In addition, emotional labor is a job stressor that leads to burnout [27]. Among three emotional labor strategies, SA is more likely to cause emotional exhaustion due to the effort required to fake or suppress negative emotions and consistently produces emotional exhaustion that results in diminished well-being [27,56]. Existing research has confirmed the relationship between emotional labor and burnout among teachers, nurses, dental hygienists, and firefighters all over the world [55,57,58,59].

As our results demonstrate, EI affects three emotional labor strategies in different ways. EI was negatively associated with SA and positively associated with GE. Interestingly, EI was not associated with DA. Our findings are consistent with those of a study by Psilopanagioti et al., which showed a negative correlation between EI and SA [29]. These results could be explained using the COR theory by Hobfoll [23]. EI is seen as a valuable resource for identifying the emotional states of self and others and using emotions to guide thoughts and behaviors [60]. Particularly, for athletes in particular, high EI is often associated with athletic performance [13]. For SA in emotional labor, repressing real emotions requires significant cognitive resources. For example, when a teammate makes a simple mistake, other teammates themselves might engage in SA by hiding negative expressions or gestures and continuing to offer encouragement. In contrast, using GE as an emotional labor strategy to express emotions consumes fewer resources [61]. Such unconscious GEs of real emotion reduce the likelihood of emotional dissonance, thereby decreasing the likelihood of burnout [35]. When athletes are in a high-pressure environment for a long time, their prolonged SA and less GE undoubtedly increase their emotional resources, leading to long-term burnout [62]. Commonly, people with lower EI tend to be more likely to employ SA, which indicates that people with lower EI have less ability to control and regulate their emotions and are unable to resolve negative emotions in time. Hence, they will only adjust their emotions in their expressions and not truly adjust the actual emotions inside [28]. Sliter et al. conducted a study to investigate the associations of age, emotional labor strategies, and EI in service occupations. The age of the study participants ranged from 18 to 68 years. The study found that EI partially affects the association between age and emotional labor strategies [31]. This may also help to explain the findings as to why adolescents may be more prone to use SA strategies.

Additionally, our findings indicated that SA and GE partially mediate the associations between EI and burnout. Similarly, Liu et al. found in a cross-sectional study that emotional labor strategies can be effective in reducing burnout through EI [32]. Another cross-sectional study involving 322 American high school coaches also reached the same conclusion. The study shows that coaches with high EI actively strive to experience and express appropriate emotions naturally (i.e., GE) instead of resorting to superficial strategies such as suppressing or faking emotions (i.e., SA) [11]. Our findings suggest that interventions targeting emotional labor strategies may be effective in reducing burnout in adolescent athletes. Studies have shown that athletes can improve EI by participating in mindfulness interventions or training sessions [63,64]. Additionally, research by Wagstaff et al. conducted educational workshops and one-to-one coaching sessions for national coaches and team managers, club coaches, and national talent academy athletes. The results indicated that the intervention group was effective in promoting the use of more adaptive and reducing maladaptive emotional regulation strategies [65]. Therefore, adolescent athletes could receive training on EI and emotional labor strategies to improve their ability to manage their emotions in high-pressure environments, thereby promoting the use of effective strategies for emotional labor and ultimately reducing burnout.

Our study could be one of the first to focus on adolescent basketball players to investigate the relationship of EI and burnout with emotional labor strategies. However, it still has several limitations. Firstly, this study is a cross-sectional design. The observed results do not reflect causality. Future studies could use a longitudinal design to examine the causal relationships between EI, emotional labor strategies, and burnout over time. Secondly, the original data was obtained through a questionnaire survey. The self-reported measurement may have an impact on the model results of constructing relationships between variables. Thirdly, the study in this article was conducted on adolescent basketball players, and the results obtained may not be applicable to other sports or individual projects. Future studies could consider distributing the questionnaire among a wider group of athletes. Lastly, gender may be a potential moderating factor. However, limited by the sample size in this study, it was insufficient to support analyzing gender differences in the mediating model.

Despite the limitations, the findings underscore the utility of EI training for athletes to manage emotional demands and reduce burnout risks. Practical applications could include EI workshops focusing on self-awareness and emotion regulation, as well as training in emotional labor strategies that encourage GE over SA. For future research, investigating the effectiveness of such interventions in longitudinal studies would be valuable. Additionally, examining the role of organizational support and cultural differences in mediating the EI-burnout link could provide further insights for sports psychologists and coaches in diverse settings.

## 5. Conclusions

Our findings indicated that EI is significantly linked to reduced burnout among adolescent basketball players, with SA and GE emerging as key mediators. These findings underscore the potential of targeted EI and emotional labor strategy training to mitigate burnout in basketball training.

## Figures and Tables

**Figure 1 sports-12-00266-f001:**
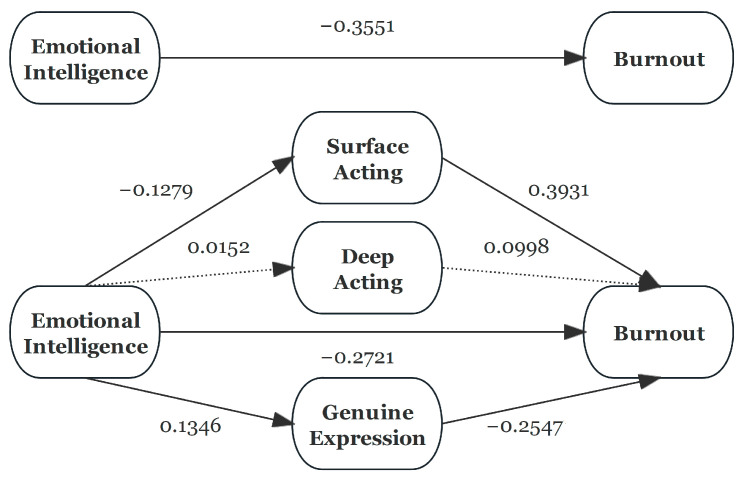
Mediation model of three emotional labor strategies between EI and burnout.

**Table 1 sports-12-00266-t001:** Demographic characteristics.

Variables	N = 260
Gender	
*Male*	70 (26.9%)
*Female*	190 (73.1%)
Age	15.10 ± 1.95
YoT	3.93 ± 2.48
Level	
*First-class*	65 (25.0%)
*Second-class*	44 (16.9%)
*None*	151 (58.1%)

Abbreviations: YoT—Years of Training.

**Table 2 sports-12-00266-t002:** CFA of default model and model with a single common factor.

Model	CFI	GFI	NFI	RMR	RMSEA	χ^2^/df
Default	0.666	0.620	0.591	0.170	0.099	2711.908/769 = 3.527
One single common factor	0.471	0.443	0.419	0.203	0.123	3853.917/779 = 4.947

**Table 3 sports-12-00266-t003:** Correlation between variables.

	Gender	Age	YoT	Level	EI	SA	DA	GE	Burnout
Gender	1								
Age	0.110	1							
YoT	0.207 **	0.558 **	1						
Level	0.376 **	0.620 **	0.563 **	1					
EI	−0.254 **	−0.051	0.097	−0.072	1				
SA	0.352 **	0.228 **	0.246 **	0.340 **	−0.169 **	1			
DA	0.298 **	0.143 *	0.170 **	0.193 **	0.019	0.614 **	1		
GE	0.098	0.045	0.068	0.061	0.168 **	0.420 **	0.506 **	1	
Burnout	0.281 **	0.302 **	0.282 **	0.418 **	−0.499 **	0.430 **	0.215 **	−0.118	1

Abbreviations: YoT—Years of Training; EI—Emotional Intelligence; SA—Surface Acting; DA—Deep Acting; GE—Genuine Expression. * *p* < 0.05; ** *p* < 0.01.

**Table 4 sports-12-00266-t004:** Independent samples *t*-test of gender difference.

Variables	Male	Female	Male-Female	*t*	*p*
EI	4.613	4.072	0.540	4.223	0
SA	2.579	3.145	−0.566	−6.037	0
DA	2.710	3.209	−0.499	−4.339	0
GE	3.148	3.314	−0.166	−1.364	0.176
Burnout	1.841	2.267	−0.426	−5.580	0

Abbreviations: EI—Emotional Intelligence; SA—Surface Acting; DA—Deep Acting; GE—Genuine Expression.

**Table 5 sports-12-00266-t005:** Associations of EI and emotional labor strategies with burnout.

Model	Variables	β	*t*	*p*	LLCI	ULCI	VIF	R^2^	ΔR^2^	F
Model 1								0.2	0.2	15.932
	Gender	0.157	2.559	0.011	0.055	0.421	1.202			
	Age	0.086	1.127	0.261	−0.022	0.082	1.873			
	YoT	0.044	0.611	0.542	−0.026	0.050	1.638			
	Level	−0.281	−3.454	0.001	−0.348	−0.095	2.103			
Model 2								0.414	0.214	35.940
	Gender	0.018	0.328	0.743	−0.136	0.190	1.293			
	Age	0.013	0.197	0.844	−0.040	0.049	1.898			
	YoT	0.164	2.614	0.009	0.011	0.078	1.705			
	Level	−0.276	−3.957	0	−0.326	−0.109	2.103			
	EI	−0.489	−9.643	0	−0.42	−0.277	1.117			
Model 3								0.509	0.095	32.540
	Gender	−0.052	−0.994	0.321	−0.235	0.077	1.405			
	Age	0.001	0.023	0.982	−0.041	0.042	1.906			
	YoT	0.130	2.236	0.026	0.004	0.066	1.718			
	Level	−0.223	−3.435	0.001	−0.277	−0.075	2.154			
	EI	−0.416	−8.477	0	−0.365	−0.228	1.232			
	SA	0.307	4.957	0	0.174	0.404	1.965			
	DA	0.112	1.844	0.066	−0.007	0.209	1.890			
	GE	−0.252	−4.684	0	−0.319	−0.130	1.474			

Abbreviations: YoT—Years of Training; EI—Emotional Intelligence; SA—Surface Acting; DA—Deep Acting; GE—Genuine Expression.

**Table 6 sports-12-00266-t006:** The mediating effect of emotional labor.

Outcome Variable	Predictive Variable	Fitting Indicators			Coefficients					
		R	R^2^	F	β	SE	*t*	*p*	LLCI	ULCI
Burnout		0.4985	0.2485	85.3155						
	EI				−0.3551	0.0384	−9.2366	0	−0.4308	−0.2794
SA		0.1689	0.0285	7.5736						
	EI				−0.1279	0.0465	−2.7520	0.0063	−0.2195	−0.0364
DA		0.0193	0.0004	0.0963						
	EI				0.0152	0.0491	0.3104	0.7565	−0.0815	0.1119
GE		0.1684	0.0284	7.5328						
	EI				0.1346	0.0490	2.7446	0.0065	0.0380	0.2311
Burnout		0.6542	0.4279	47.6911						
	EI				−0.2721	0.0356	−7.6379	0	−0.3422	−0.2019
	SA				0.3931	0.0593	6.6267	0	0.2763	0.5099
	DA				0.0998	0.0579	1.7234	0.0860	−0.0142	0.2139
	GE				−0.2547	0.0510	−4.9964	0	−0.3551	−0.1543

Abbreviations: EI—Emotional Intelligence; SA—Surface Acting; DA—Deep Acting; GE—Genuine Expression.

**Table 7 sports-12-00266-t007:** Direct, indirect, and total effect.

Path		Effect	BootSE	BootLLCI	BootULCI	Percentage
Direct (c’)		−0.2721	0.0356	−0.3422	−0.2019	76.63%
Indirect	EL (a × b)	−0.0830	0.0231	−0.1266	−0.0387	23.37%
	SA (a1 × b1)	−0.0503	0.0205	−0.0918	−0.0101	14.17%
	DA (a2 × b2)	0.0015	0.0067	−0.0128	0.0159	−0.42%
	GE (a3 × b3)	−0.0343	0.0161	−0.0690	−0.0058	9.66%
Total (c)		−0.3551	0.0384	−0.4308	−0.2794	100%

Abbreviations: EL—Emotional Labor; SA—Surface Acting; DA—Deep Acting; GE—Genuine Expression.

## Data Availability

The data are not publicly available due to containing information that could compromise the privacy of research participants.

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
