# Peer review of "Emotional Intelligence and Burnout among Adolescent Basketball Players: The Mediating Effect of Emotional Labor"

_sports, 2024, doi:10.3390/sports12100266_

Round 1
Reviewer 1 Report
Comments and Suggestions for Authors
Thank you for the opportunity to review this manuscript, which examines the relationship between emotional intelligence, emotional labor strategies, and burnout in adolescent basketball players. The study is relevant and contributes valuable insights to sports psychology. However, I believe that certain areas require further detail and clarification. Below I offer specific suggestions for improvement. Best of luck to the authors in revising their work.
Introduction
1. The introduction talks about emotional labor strategies (surface acting, deep acting, and genuine expression), but it doesn’t fully explain why these particular variables were chosen to study in the context of athlete burnout. In contrast, the connection between emotional intelligence (EI) and burnout is better established within the sports context.
2. While the introduction mentions that EI influences emotional labor strategies and burnout, it doesn’t clearly explain the theoretical or practical mechanisms by which EI might impact these strategies in a sports setting.
3. The introduction relies heavily on literature from non-sports fields. Aren’t there relevant studies in sports that could be cited?
4. The novelty of the study isn’t strongly established. There’s a need for a clearer rationale for why this research is necessary and worth publishing. Highlight what is currently unknown or insufficiently explored, and how this study aims to fill those gaps.
Materials and Methods
5. Which sampling technique was used?
6. Please describe the criteria for determining when a questionnaire was considered invalid.
7. The manuscript mentions a "mixed total scale" was used to investigate emotional labor strategies but doesn’t clearly explain what this mixed scale includes.
8. What are the Cronbach’s alphas for each of the three emotional labor strategies in your study?
9. The statistical analysis section is brief and doesn’t provide enough detail on how the mediating effect of emotional labor strategies was tested. Please expand on this.
Results
10. To strengthen your results, consider including common fit indices like CFI, TLI, RMSEA, and SRMR for your mediation analysis, if possible.
11. When you mention that EI is negatively correlated with gender, it’s unclear what you mean. Are males or females showing higher EI? Please specify.
Discussion
12. The discussion touches on practical implications, such as training athletes to manage emotions, but these suggestions are underdeveloped and not sufficiently grounded in the study’s findings.
13. The use of only self-report measures might have biased the findings, so this should be mentioned in the limitations.
14. Instead of just suggesting longitudinal research for future studies, try to offer more specific ideas for future research directions.
Reviewer 2 Report
Comments and Suggestions for Authors
I am pleased to have the opportunity to review your manuscript titled "Emotional Intelligence and Burnout Among Adolescent Basketball Players: The Mediating Effect of Emotional Labor." This is an important and valuable topic, as burnout among adolescent athletes is a significant issue that warrants further exploration. Your focus on emotional intelligence and the role of emotional labor strategies provides meaningful insights that could contribute to the well-being and performance of young athletes.
While your study holds promise and offers valuable perspectives, I have identified several areas that require further revision and improvement. I look forward to working together to enhance the quality of your manuscript.
1. The introduction does not sufficiently highlight gaps in the existing literature that your study aims to address. Although the manuscript briefly mentions previous research on EI, emotional labor, and burnout in various professions, it lacks a discussion on the specific gap this study fills in the context of adolescent basketball players. Providing a more detailed rationale for why this study is necessary—especially in the specific context of adolescent athletes and team sports like basketball—would strengthen the justification for the research. Identifying the unique aspects of emotional labor in youth sports, as compared to other fields, would also add depth to your argument and provide a stronger foundation for your research objectives.
2. The introduction section introduces key concepts like emotional intelligence, emotional labor, and burnout, but it lacks a clear theoretical framework that ties these concepts together. To strengthen the introduction, it would be beneficial to include a theoretical model or framework that illustrates how emotional intelligence and emotional labor strategies interact to influence burnout. This framework would not only guide the study but also help in the formulation of clear, testable hypotheses. Presenting a visual model or explicitly stating the hypotheses based on existing theories would provide a clearer direction for the research and enhance the readers' understanding of the study's approach.
3. The manuscript currently lacks a comprehensive literature review section, which is crucial for establishing the context and foundation of the study. A literature review would provide a more detailed examination of previous research related to emotional intelligence, emotional labor strategies, and burnout, especially in the context of adolescent athletes and sports.
4. Although the manuscript provides a good description of the measurement tools used, such as the Wong and Law Emotional Intelligence Scale (WLEIS) and the Athlete Burnout Questionnaire (ABQ), it lacks detail about their cultural adaptation and validity in the specific context of Chinese adolescent athletes. It is crucial to mention whether the versions of these scales have been validated in similar cultural and age-specific populations.
5. The manuscript mentions the use of Harman’s single-factor test to check for common method bias, with results indicating no serious bias. However, this test alone might not be sufficient to rule out common method bias comprehensively. It would be advisable to employ additional methods, such as confirmatory factor analysis (CFA), to further assess the presence of common method variance. This would provide more robust evidence that common method bias did not unduly influence the findings.
6. The discussion provides a summary of the findings but lacks depth in explaining the underlying mechanisms by which emotional intelligence (EI) and emotional labor strategies influence burnout. While the Conservation of Resources (COR) theory is mentioned to explain why surface acting (SA) leads to burnout and genuine expression (GE) reduces it, the discussion could benefit from a more detailed explanation of how EI specifically interacts with these emotional labor strategies.
7. The discussion does a good job of referencing studies from other fields (e.g., healthcare and education) that show similar relationships between EI and burnout. However, there is limited comparison with previous research specifically within sports psychology or among adolescent athletes. The discussion could be strengthened by referencing more studies that have examined EI, emotional labor, and burnout within athletic or sports settings.
8. While the discussion briefly suggests that interventions targeting emotional labor strategies may be effective in reducing burnout, it lacks specific recommendations for practical applications. The discussion should provide more concrete suggestions on how coaches, sports psychologists, and trainers can implement these findings in real-world settings.
Comments on the Quality of English LanguageThe quality of the English language in this manuscript is generally acceptable, but there are some areas that need improvement for clarity and readability. Some sentences are complex or awkwardly phrased, which may hinder understanding. I suggest that the authors consider revising these sections for clarity and conciseness.
Round 2
Reviewer 1 Report
Comments and Suggestions for Authors
The authors have adequately resolved all of my concerns.
Author Response
Thank you very much for your positive review of our revised manuscript. We are delighted to hear that you are satisfied with the changes we have made in response to your previous comments and that you have no further queries regarding the current version of the manuscript.
Your acknowledgment of our efforts to address the issues raised in your initial review is greatly appreciated. It has been our priority to thoroughly resolve all concerns and to enhance the quality and clarity of our research presentation. We are pleased that you found our revisions adequate and that the manuscript now meets the standards you expect.
We are also grateful for the time and attention you have given to our work. The constructive feedback you provided during the earlier stages of the review process was instrumental in guiding us toward this improved version of our manuscript.
Once again, thank you for your support and for the opportunity to contribute to the journal. We look forward to any further guidance you may have as our manuscript progresses through the publication process.
Reviewer 2 Report
Comments and Suggestions for Authors
Thank you for your thorough revisions. You have addressed the key points from the initial review very well, particularly in expanding the introduction, adding a comprehensive literature review, and incorporating a theoretical framework. The inclusion of cultural validation for the measurement tools and the additional statistical analyses, such as the confirmatory factor analysis, have greatly strengthened the methodological rigor of the study.
Additionally, the clearer explanation of how emotional intelligence interacts with emotional labor strategies and the more detailed discussion of practical applications for coaches and sports psychologists are significant improvements.
Overall, the manuscript is much clearer and more robust, and I appreciate your efforts in enhancing the clarity and quality of the paper. A few minor points for further consideration include ensuring that the newly added sections flow smoothly with the rest of the manuscript and reviewing the language for any remaining awkward phrasing. These adjustments will further improve readability and coherence.
Comments on the Quality of English LanguageOverall, the quality of English in the manuscript is acceptable and the key points are communicated clearly. However, there are a few instances where sentence structure could be simplified for better readability, and some phrases may benefit from rephrasing to improve clarity. I recommend a final language editing to address minor grammatical issues and ensure that the flow of the text is smooth throughout the manuscript.
Author Response
Thank you very much for your constructive feedback and the positive evaluation of our revised manuscript. We are pleased that you found the revisions satisfactory and that the manuscript has significantly improved in clarity and rigor.
Comment 1: Thank you for your thorough revisions. You have addressed the key points from the initial review very well, particularly in expanding the introduction, adding a comprehensive literature review, and incorporating a theoretical framework. The inclusion of cultural validation for the measurement tools and the additional statistical analyses, such as the confirmatory factor analysis, have greatly strengthened the methodological rigor of the study.
Additionally, the clearer explanation of how emotional intelligence interacts with emotional labor strategies and the more detailed discussion of practical applications for coaches and sports psychologists are significant improvements.
Overall, the manuscript is much clearer and more robust, and I appreciate your efforts in enhancing the clarity and quality of the paper. A few minor points for further consideration include ensuring that the newly added sections flow smoothly with the rest of the manuscript and reviewing the language for any remaining awkward phrasing. These adjustments will further improve readability and coherence.
Response 1: We greatly appreciate your commendation on our expanded introduction, comprehensive literature review, and the incorporation of a theoretical framework. We have taken your suggestion to ensure that the newly added sections flow smoothly with the rest of the manuscript. To achieve this, we have carefully restructured the document, paying particular attention to the transitions between the sections. We have also revisited each section to ensure that the language is consistent and that the narrative progresses logically from one part to the next.
Furthermore, we acknowledge the importance of methodological rigor. The inclusion of cultural validation for our measurement tools and additional statistical analyses, such as confirmatory factor analysis, was a critical step that we undertook to address the initial review comments. We believe these additions have indeed strengthened our methodology and the overall quality of the research.
We also understand the significance of clearly explaining how emotional intelligence interacts with emotional labor strategies. To this end, we have refined our discussion to provide a more detailed and accessible explanation of these interactions. Similarly, we have expanded upon the practical applications for coaches and sports psychologists to ensure that the section is not only informative but also actionable.
Comment 2: Overall, the quality of English in the manuscript is acceptable and the key points are communicated clearly. However, there are a few instances where sentence structure could be simplified for better readability, and some phrases may benefit from rephrasing to improve clarity. I recommend a final language editing to address minor grammatical issues and ensure that the flow of the text is smooth throughout the manuscript.
Response 2: We agree with your observation regarding the English language quality of our manuscript. While the key points are communicated clearly, we recognize that there is always room for improvement when it comes to readability and grammatical accuracy. To address this, we have engaged a professional language editor to review and refine our manuscript. This final language editing pass has focused on simplifying sentence structures, rephrasing certain phrases, and correcting any minor grammatical issues. We believe these changes have resulted in a manuscript that flows smoothly and is accessible to a broader readership.
We are confident that these final adjustments will address your concerns and further enhance the quality of our work. Once again, we thank you for your insightful comments and for the time you have dedicated to reviewing our manuscript. We have endeavored to address each of your points with the utmost diligence and appreciate your guidance in helping us improve our paper.